# Development of the ADFICE_IT clinical decision support system to assist deprescribing of fall-risk increasing drugs: A user-centered design approach

Sara S. Groos[1,2]*, Kelly K. de Wildt[1,2], Bob van de Loo[1,2,3], Annemiek J. Linn[4], Stephanie Medlock[2,5,6], Kendrick M. Shaw[6,7,8], Eric K. Herman[9], Lotta J. Seppala[1,2], Kim J. Ploegmakers[1,2], Natasja M. van Schoor[2,3], Julia C. M. van Weert[4], Nathalie van der Velde[1,2]

1 Internal Medicine, Section of Geriatric Medicine, Amsterdam UMC Location University of Amsterdam, Amsterdam, the Netherlands, 2 Amsterdam Public Health Research Institute, Amsterdam, the Netherlands, 3 Epidemiology and Data Science, Amsterdam UMC Location Vrije Universiteit Amsterdam, Amsterdam, the Netherlands, 4 Amsterdam School of Communication Research/ASCoR, University of Amsterdam, Amsterdam, the Netherlands, 5 Department of Medical Informatics, Amsterdam UMC Location University of Amsterdam, Amsterdam, the Netherlands, 6 Stichting Open Electronics Lab, Maarssen, The Netherlands, 7 Department of Anesthesia, Critical Care and Pain Medicine, Massachusetts General Hospital, Boston, MA, United States of America, 8 Harvard Medical School, Boston, MA, United States of America, 9 Commons Caretakers BV, Amsterdam, The Netherlands

* s.s.groos@amsterdamumc.nl

**Data Availability Statement:** All relevant data are within the manuscript.

## Abstract

### Introduction

Deprescribing fall-risk increasing drugs (FRIDs) is promising for reducing the risk of falling in older adults. Applying appropriate deprescribing in practice can be difficult due to the outcome uncertainties associated with stopping FRIDs. The ADFICE_IT intervention addresses this complexity with a clinical decision support system (CDSS) that facilitates optimum deprescribing of FRIDs by using a fall-risk prediction model, aggregation of deprescribing guidelines, and joint medication management.

### Methods

The development process of the CDSS is described in this paper. Development followed a user-centered design approach in which users and experts were involved throughout each phase. In phase I, a prototype of the CDSS was developed which involved a literature and systematic review, European survey (n = 581), and semi-structured interviews with clinicians (n = 19), as well as the aggregation and testing of deprescribing guidelines and the development of the fall-risk prediction model. In phase II, the feasibility of the CDSS was tested by means of two usability testing rounds with users (n = 11).

**Funding:** The ADFICE_IT project is supported by funding from the Netherlands Organization for Health Research and Development (ZonMw, Grant 848017004), The Hague and the Amsterdams Universiteitsfonds: Gepersonaliseerde Medicatieaanpassing bij Oudere Vallers. KMS was supported by US NIH grant T32-GM007592. ZonMw: https://www.zonmw.nl/en Amsterdams Universiteitsfonds: https://www.auf.nl/en NIH: https://www.nih.gov/grants-funding The funders had no role in study design, data collection and analysis, decision to publish, or preparation of the manuscript.

**Competing interests:** The authors have declared that no competing interests exist.

## Results

The final CDSS consists of five web pages. A connection between the Electronic Health Record allows for the retrieval of patient data into the CDSS. Key design requirements for the CDSS include easy-to-use features for fast-paced clinical environments, actionable deprescribing recommendations, information transparency, and visualization of the patient's fall-risk estimation. Key elements for the software include a modular architecture, open source, and good security.

## Conclusion

The ADFICE_IT CDSS supports physicians in deprescribing FRIDs optimally to prevent falls in older patients. Due to continuous user and expert involvement, each new feedback round led to an improved version of the system. Currently, a cluster-randomized controlled trial with process evaluation at hospitals in the Netherlands is being conducted to test the effect of the CDSS on falls. The trial is registered with ClinicalTrials.gov (date; 7-7-2022, identifier: NCT05449470).

## Introduction

Falls and fall-related injuries among older adults are a growing major public health problem [1]. In 2017, 11.7 million older adults in Western Europe requested medical treatment for an injury, of which 8.4 million were fall-related [2]. Injurious falls may result in admission to long term care, loss of independence, reduced mobility, fear of falling, and social isolation, significantly reducing the quality of life for older adults [3–5]. As a result, falls place a significant financial burden on healthcare systems. In Western countries, it is estimated that up to 1.5 percent of the total healthcare expenditures are attributed to fall-related medical care costs [3, 6].

A prominent risk factor for falls in older adults is the use of certain medication classes known as fall-risk increasing drugs (FRIDs). The use or combined use of FRIDs, such as psychotropics and cardiovascular drugs, is associated with adverse effects including orthostatic hypertension, syncope, sedation, and dizziness that can cause accidental falls in older adults [7–10]. Therefore, appropriate deprescribing of FRIDs is recommended for lowering an older adult's risk of incurring a medication-related fall [11]. Nevertheless, previous studies suggest that deprescribing approaches have yet to be optimized, as current approaches do not sufficiently address the complexity of FRIDs deprescribing for physicians [12, 13].

Deprescribing FRIDs is highly complex due to healthcare professional, patient, cultural and organizational reasons. For example first, physicians themselves may perceive difficulties with which, how and when a FRID or combination of FRIDs should be safely deprescribed. This results in the reluctancy to deprescribe, a phenomena that is especially prominent in the treatment of older patients with polypharmacy and multimorbidity [14, 15]. Second, the unique risk profiles of these patients can greatly increase the complexity of FRIDs deprescribing due to the weighing of competing clinical practice guidelines [15, 16]. As a result, physicians may have a tendency to overestimate the expected benefits of medications (i.e., effective treatment of the chronic condition) and underestimate the potential harms (i.e., an injurious fall) in these patients [17]. Third, physicians may also experience deprescribing reluctance from the patient. Research suggests that low medication-related knowledge in geriatric patients, such as the management of medications, can negatively influence appropriate deprescribing [18].

One way to better support physicians in the deprescribing of FRIDs is by means of a clinical decision support system (CDSS). Such a system has the ability to link individual patient characteristics to a computerized clinical knowledge base which uses information from guidelines to generate patient-specific recommendations back to the physician. These recommendations can subsequently be discussed together with the patient [19, 20]. In the context of FRIDs, a CDSS has the potential to support physicians through a structured deprescribing approach by first signaling which medications pose a risk to the individual patient, and subsequently advising how and when to safely deprescribe each medication (i.e., through guideline integration). In turn, the decision to deprescribe certain medications can be discussed together with the patient to foster joint medication management between physicians and patients. Such shared decision-making has been shown to improve medication-related knowledge and medication adherence in older patients, and could thus enhance the effectiveness of fall preventive care in these patients [21].

Previous studies suggest that medication-related CDSS can improve care outcomes for older patients in a variety of contexts (e.g., inappropriate medication use by patients and polypharmacy, inappropriate prescribing by physicians, falls) [22–24]. However, these studies have not used the potential of a CDSS in deprescribing optimally, such as incorporating data-driven methods, like prediction models, in the system's clinical knowledge base [24]. Prediction models combine data for multiple risk factors in order to calculate the risk of a future outcome, and help inform subsequent decision making [25]. Thus, in the context of deprescribing optimally, such models could serve as an adjunct to decision-making by allowing the physician and patient to weigh the various treatment options on the basis of the patient's risk of falling within 12 months. Moreover, such estimates of fall risk could help patients be more aware of their risk of falling and as such motivate them to follow the advice recommended by the physician.

The acceptance of CDSSs among physicians is still hindered by a number of barriers, such as insufficient knowledge with system use, time-consuming, alert fatigue and poor integration into workflow [24, 26–28]. A possible explanation for the considerable number of usability barriers is the lack of involvement by physicians during the development of these systems. A user-centered design approach is an iterative method that involves users of a system in each stage of the development process. Such an approach has been found to enhance the ease of use and usefulness of clinical study tools–the two important determinants that can influence technology adoption by clinicians [29, 30].

The current paper aims to outline how user and expert insights were used to inform the development of a data-driven CDSS that aims to provide optimal support for physicians during the deprescribing of FRIDs by generating a personalized fall-risk estimate and guideline-based medication advice tailored to the health conditions of each individual patient. Following a user-centered design approach, we showcase how users were successfully involved in different stages of the development process to increase the system's acceptance and adoption in its intended clinical care setting later on. This CDSS is part of the AD*F*ICE_IT (**A**lerting on adverse **D**rug reactions: **F**alls prevention **I**mprovement through developing a **C**omputerized clinical support system: **E**ffectiveness of **I**ndividualized medica**T**ion withdrawal) intervention, a novel deprescribing intervention that aims to prevent medication-related falls in older adults by means of a CDSS for clinicians and an online portal for patients [31].

## Materials and methods

### Ethics statement

The Medical Ethics Research Committee of the Amsterdam University Medical Center (Amsterdam UMC, location University of Amsterdam; W19_310 # 19.368) declared that the

Medical Research Involving Human Subjects Act did not apply to this study. All study participants gave written informed consent prior to data collection.

## Medical research council framework

The overall AD*F*ICE_IT intervention is developed and evaluated following the four phases of the Medical Research Council framework (MRC): (I) the development, (II) the feasibility, (III) the implementation, and (IV) the evaluation phase. The MRC is a guiding theoretical framework for developing, pilot-testing, implementing and evaluating complex health interventions [32]. This paper summarizes the (I) development and (II) feasibility phases for our CDSS. An overview of the studies conducted in these phases is provided in Table 1.

## Phase I: Development

The aim of this phase was to cultivate a robust (theoretical) understanding about how to develop the CDSS components [30], which consisted of the (1) development of the user interface, (2) development of the clinical knowledge base, (3) development of the prediction model, and (4) development of the software. A detailed description of the methods for the systematic

**Table 1. The development process of the AD*F*ICE_IT CDSS.**

| MRC Phase I: Development | |
| --- | --- |
| **User interface** | |
| Dec. 1 2018 –July 15 2019 | **European survey**: Barriers and facilitators in using a CDSS for fall risk management for older adults [34] |
| Dec. 17 2019 –Jan. 3 2020 | **Semi-structured interviews***: Clinician needs for effective implementation and trustworthy decision making |
| 2020 | **Literature review***: Risk communication needs of physicians |
| 2020 | **Systematic review**: Barriers and facilitators influencing medication-related CDSS acceptance according to clinicians [33] |
| **Knowledge base** | |
| Used latest edition 2017 | **Dutch fall guideline**: Effect of medication on fall risk in older adults [37] |
| Feb. 5 2019 –Feb. 28 2020 | **Modified Delphi study**: STOPPFall (Screening Tool of Older Persons Prescriptions in older adults with high fall risk): a Delphi study by the EuGMS Task and Finish Group on FRIDs [35] |
| 2020–2021 | **Logical elements rule method***: Formalizing clinical rules for the CDSS [38] |
| **Reasoning Engine and Software** | |
| 2021–2022 | **Agile methodology using test-driven development***: Unit tests, acceptance tests, verification, and validation testing |
| **Prediction model** | |
| 2020–2021 | **Development**: The AD*F*ICE_IT models for predicting falls and recurrent falls in community-dwelling older adults: Pooled analyses of European cohorts with special attention to medication [36] |
| MRC Phase II: Feasibility | |
| **CDSS prototype 1** | |
| Sept. 10–24 2020 | **Usability testing round 1***: Identification of usability problems and adjustments to prototype |
| **CDSS final version** | |
| June 28 –July 18 2021 | **Usability testing round 2***: Identification of remaining usability problems and development of final version |

* The methods and results are described in detail in this study.

review, European online survey, modified Delphi study, and development of the fall-risk prediction model have been published elsewhere by the research team [33–36].

**Development of the user interface.** To develop the user interface, a literature review on risk communication needs and a systematic review on barriers and facilitators were carried out, and extended with empirical research by means of a European online survey (n = 581) distributed among physicians and semi-structured interviews with clinicians (n = 19) from different hospitals in the Netherlands. The literature review assessed the risk communication needs of physicians. References for the review were retrieved from Google Scholar using the following search terms: "doctor," "physician," "healthcare provider," "clinical decision support system," "clinical support system," "satisfaction," "preferences," "usability," "user centered design," "risk information," "risk communication," "medication," "drug," "drug on drug." References were excluded if (1) the sample did not compromise of physicians; (2) the study did not discuss a medication-related CDSS; (3) the study was not published between 2014 and 2020. The systematic review assessed barriers and facilitators to CDSS use [33]. The latter barriers and facilitators to use analysis was expanded on in the survey (n = 581), which was distributed among European physicians, nurse practitioners, and physician assistants who in their clinical practice (primary, secondary, and tertiary care) see older adults at risk of falling [34]. In-person, semi-structured interviews with clinicians were conducted to assess the needs for effective implementation and trustworthy decision making with the help of a topic-list developed for this study. For example, clinicians were asked how the system should communicate a patient's fall risk; or how the system can best support clinicians in making informed decisions about whether or not to deprescribe. Clinicians (n = 19) were eligible to participate if they (1) are working at a hospital in the Netherlands where the AD*F*ICE_IT intervention was set to be implemented and (2) regularly perform a multifactorial fall risk assessment in older adults at risk of falling in an (geriatric) outpatient setting. All interviews were conducted in Dutch, voice-recorded, and transcribed verbatim. The analysis of these studies guided the development of the user interface of the CDSS, such as the system's functionality, design, and content composition.

**Development of the knowledge base.** The STOPPFall (Screening Tool of Older Persons Prescriptions in older adults with high fall risk) tool [35] and the Dutch fall guideline [37] were used to identify relevant FRIDs in different medication classes. STOPPFall entails a FRIDs list with accompanying guidelines for deprescribing, and was constructed using a modified Delphi technique through consensus effort with 24 panelists from 13 European countries [35]. Both STOPPFall and the Dutch fall guideline form the basis of the system's clinical knowledge base. To extend the knowledge base specific deprescribing advice for each class of FRIDs was identified from over 30 guidelines, and formalized using an adaptation of the Logical Elements Rule Method (LERM). LERM is a validated method for formalizing clinical rules for decision support (e.g., a CDSS). The method follows a step-by-step approach in which clinical rules are formulated by an informatics knowledge expert, and close collaboration from a clinical expert is sought throughout rule formalization to ensure that the intent of the guidelines are maintained [38]. Specifically, rather than using conjunctive normal form as specified in LERM, the criteria for each rule were formalized as medications to include (e.g., a class of FRIDs), medications to exclude (e.g., non-FRIDs within that class), and conditions (e.g. diagnoses or laboratory values that modify the advice).

**Development of the reasoning engine and software.** Reasoning engine and software development followed an agile methodology, including using test-driven development. Both unit tests (which test specific functions individually) and acceptance tests (which test larger parts of the software as a whole) were used. The software was developed in Node JS, using Express and MariaDB. Jest and TestCafe are used for testing [39–43]. For transparency, all

software was developed as open-source software. Experienced developers were involved in creating the software architecture and establishing the development principles. Given the low expected load, clarity of code was prioritized over efficiency and scalability. Importantly, good traceability between the specification for the knowledge base and the implementation of these rules in the software was maintained throughout development, including the provision of evidence behind the recommendations to the end user.

Verification and validation testing were also conducted. Regarding verification, test cases (with specified input and expected output) were developed for each rule in the specification. If any output was not as expected, the corresponding part of the logic was checked for errors and, if needed, corrected. This was repeated until a 100% pass rate was achieved. These tests were added to the test suite and are run every time a change is made to the software. Validation testing took place in two rounds. First, hypothetical patient cases were constructed based on the specification. Specifically, cases were designed such that each text that the CDSS can produce appeared for at least one patient. The output was reviewed by NvdV to confirm that the advice was clinically sound and reflected the intent of the underlying guidelines. Any identified problems were corrected and the test was repeated. Second, informed consent was obtained from 10 patients and data from these patients was entered into the CDSS. NvdV and an expert in geriatric pharmacy reviewed the cases and commented if they did not agree with the advice. Since clinicians can have differing opinions on the same case, we aimed for 90% agreement with these recommendations.

**Development of the prediction model.** To enable the estimation of a patient's risk of falling within 12 months, a fall-risk prediction model was developed. The development of the model is described in detail in Van de Loo, Seppala et al. [36], but briefly the model was developed based on a harmonized dataset comprised of two Dutch and one German cohort studies of community-dwelling older adults (65+), namely Longitudinal Aging Study Amsterdam (LASA), B-vitamins for Prevention of Osteoporotic Fractures study (B-PROOF), and Activity and Function in the Elderly in Ulm Study (ActiFE Ulm). For background, LASA is a prospective cohort study to determine predictors and consequences of aging in older adults in the Netherlands [44]. B-PROOF is a Dutch randomized, double-blind, placebo-controlled trial among older adults with an elevated homocysteine concentration [45]. ActiFE Ulm is a population-based cohort study among community dwelling older adults in Ulm and nearby regions in southwestern Germany [46]. From these cohorts, 5722 older adults (65+) for whom medication and follow-up data were recorded were included in the development of the prediction model. This prediction model comprises a part of the CDSS's knowledge base. The outcome variable was defined as any fall (one or more falls) within a one-year follow-up. Candidate predictors were selected based on previously reported risk factors for falls that are easily obtained from the Electronic Health Record (EHR), such as sociodemographic variables; measures of emotional, cognitive and physical functioning; self-reported chronic conditions; variables related to lifestyle; biomarkers; and use of certain medications. Logistic regression with backward elimination was used to develop the model. The prediction model was internally validated using an internal-external cross-validation procedure, in which the performance of the model was tested in each of the development cohorts separately following Steyerberg and Harrel [47]. Performance was assessed using the C-statistic, whereby a C-statistic value of 0.5 indicates no discrimination and a value of 1 indicates perfect discrimination. Calibration plots were used to assess the agreement between predicted risks and observed outcomes. The prediction model was externally validated in data from geriatric outpatients (refer to Van de Loo, Heymans et al. for detailed methods) [48].

## Phase II: Feasibility

**Usability testing of the CDSS prototype.** The feasibility phase assessed whether the prototype of the CDSS had any usability problems that needed to be fixed prior to developing the final version of the CDSS. This was achieved through two usability testing rounds with geriatric physicians (n = 11) from Dutch hospitals (i.e., the primary users of the system). Usability research argues that five participants are sufficient for detecting 80% of a product's usability problems [49, 50]. Each physician was presented with three hypothetical patient cases that varied in fall risk (e.g., high versus low risk). Next, physicians were asked to carry out a scripted navigation for each case, which consisted of realistic CDSS task scenarios, resembling aspects of clinical documentation (e.g., analyzing a patient's fall-risk, selecting relevant treatment options, making a referral). Throughout navigation, physicians were prompted to "think aloud" and verbalize their thought process while carrying out the tasks (i.e. using a concurrent think aloud method). Additionally, Camtasia 9, a usability software, was used to record the screen and mouse movements of each physician, including facial expressions and vocalizations (i.e., questions, expressions of confusion) [51].

Usability problems were identified for each physician session, and coded according to the Nielsen usability problems severity rating ranging from *0* = "I don't agree that this is a usability problem at all" to *4* = "Usability catastrophe: imperative to fix this before product can be released" [52]. The categorization and potential negative impact of each identified usability problem was assessed following the augmented scheme for classifying and prioritizing usability problems [53]. Next, usability problems from all sessions were merged, and for each problem the occurrence of that problem was noted. This led to an overview of usability problems ordered on both severity and occurrence, ranging from the most severe and most often occurring problems to the least severe and least occurring problems. The analysis of the results from this study led to adjustments to the CDSS, resulting in a second version of the prototype. The aforementioned procedure was repeated in the second usability study among six physicians of which three did not participate in the first usability study. This allowed us to pinpoint remaining usability issues that needed to be improved prior to developing the final version of the CDSS.

## Results

The analysis of the results from all studies guided the development of the CDSS. In this article, we report in detail the results of the literature review on risk communication needs, semi-structured interviews, software development, and usability testing rounds. The results of the systematic review on barriers and facilitators, European online survey, modified Delphi study, and development of the fall-risk prediction model are reported in detail elsewhere by the research team [33–36].

## Phase I: Development

**Results from the literature review.** The aim of the literature review was to identify the risk communication needs of physicians. A total of six articles were included. The literature suggests that physicians prefer a holistic- and patient-specific approach to communicating health-related advice [26]. This approach should include information that is actionable, directive and transparent [54–57]. When visualizing risks, information-oriented graphs were perceived as easier to interpret by physicians, and were found to enhance shared decision-making between physicians and patients [56–58]. Similarly, using a traffic-light coloring system for the presentation of risk is believed to facilitate information processing, which can lead to a more rapid assessment of risk-based information by physicians [54, 56, 58]. Additionally, favorable

effects were found on information processing outcomes (e.g., enhanced attention, reduced cognitive load) when the formatting of information and use of terminology was consistent, and when text density and visual clutter was reduced [56, 57].

**Results from the semi-structured interviews.** The aim of the interviews with clinicians was to identify the information-related needs for effective and trustworthy decision making, and the requirements for successful implementation of the CDSS in practice. A total of 19 clinicians employed by different hospitals in the Netherlands participated of which 15 were geriatric physicians followed by two nurse practitioners, a general practitioner in training, and a physical therapist. Results showed that fall risk information should be displayed in color as either a number or percentage. With regard to effective patient-physician communication, a visual graph was viewed as beneficial for the patient, including the ability to print out a patient-friendly handout. This handout should include information about the patient's personalized fall risk and treatment plan that was discussed during the consultation. Opportunities to read about the benefits and side effects of deprescribing a medication, and having access to additional information (e.g., via hyperlinks) were viewed as important in the decision making to deprescribe. Moreover, information about how a patient's fall risk is calculated (i.e., the prediction model) was perceived as vital information by physicians that would also enhance the system's credibility. Lastly, for successful implementation of the CDSS in practice, physicians stressed the importance of reducing completion time (e.g., limiting the amount of mouse clicks).

**Development of the user interface and knowledge base.** Table 2 shows the aggregated system needs of clinicians obtained from the literature review on risk communication needs, systematic review on barriers and facilitators, European online survey, and semi-structured interviews with clinicians. These key requirements were subsequently operationalized into system features used for the development of the user interface of the first CDSS prototype for usability testing. Regarding the knowledge base of the system, the final set of clinical rules covering 22 classes of FRIDs, with specific deprescribing advice based on diagnoses, lab values, and concurrent medications, including the final fall-risk prediction model were integrated into the CDSS.

The results of the prediction model are described in detail in Van de Loo, Seppala et al. [36] and Van de Loo, Heymans et al. [48]. The final prediction model consisted of the following 14

**Table 2. Aggregated needs of clinicians.**

| Key requirements for CDSS
R: Literature reviews
S: Survey
I: Semi-structured interviews |
|---|
| • Limit repeated and uninformative alerts (R, S). |
| • Include easy-to-use interactive features that cater towards physicians fast-paced work environment (R, I). |
| • Provide physicians with actionable recommendations on how a patient's fall risk can be reduced (R). |
| • Make information transparent and credible by including hyperlinks that direct physicians to information surrounding the fall-risk prediction model and other medication-related information (I). |
| • The CDSS should provide a holistic overview of the patient (e.g., comorbidities) (R). |
| • The format of the CDSS should be consistent (R). |
| • The presentation of risk should be presented in a text-format using concise and to-the-point language (e.g., using short sentences with standardized terminology) (R). |
| • Information should be presented in a systematic manner that complements the consultation workflow of physicians (R, S). |
| • A patient's fall risk should be accompanied with information-orientated graphs (e.g., bar graphs, icon arrays, pie-charts) (R, I). |
| • The information-orientated graphs should employ a traffic-light coloring system (i.e., red for high risk, yellow for medium, risk, and green for no risk) (R, I). |
| • The CDSS should be efficient in use (i.e., fast completion time, limited clicks and text-entry fields) (I, S). |

predictors: educational status (low, middle, high), depression (different validated scales were combined using z-scores), body mass index, grip strength (in kg), gait speed (in meter per second), number of functional limitations (from 0 to 5), systolic blood pressure (in mmHg), at least one fall in the previous 12 months, at least two falls in the previous 12 months, fear of falling (from 0 = not afraid to 2 = very afraid), smoking status (0 = never to 2 = current smoker), use of calcium channel blockers, use of antiepileptics, and use of drugs for urinary frequency and incontinence (see Van de Loo, Seppala et al. for harmonization guide) [36]. Performance of the prediction model was comparable to other fall-risk prediction models with a mean C-statistic value of 0.65 in the cohorts used for model development (range 0.61–0.67) [36, 59]. Calibration plots revealed good agreement between the predicted risks and observed outcomes [36]. External validation of the model showed similar performance in geriatric outpatients as in the cohorts that were used for development, with a C-statistic of 0.66 [48].

**Development of the reasoning engine and software.** Following agile development principles, development of the software started by building the smallest part that would be useful, which was the reasoning engine and an interface for verification and validation testing that showed the advice text for the doctor, checkbox options, corresponding patient-friendly text, and references all on one screen. Features were then added in order of priority. This, in combination with test-driven development, lead to a modular architecture with good separation between the connection to the EHR, the logic, and the user interface. This allowed developers to readily modify the interface based on feedback received from phase II. Development as open source software facilitated seeking and incorporating input from external expert developers (see acknowledgements). Regarding the system's security, the software is designed to be hosted within the hospital network, and dependencies are minimized to lower the security footprint and improve maintainability. The CDSS logic is kept server side to ensure that it cannot be changed from the browser, and that the information seen by the user is always consistent with the saved data. Features were added to limit data being cached in the user's browser. After verification testing and the first round of validation testing were completed with no remaining errors detected, the second round of validation testing with real patient data was conducted. The 10 patients were taking 42 different medications, triggering 50 distinct rules. This resulted in 100% agreement with these recommendations from one clinician and 95% agreement from the other, which exceeded the minimum threshold of 90% agreement.

## Phase II: Feasibility

**Results from usability testing rounds.** To identify usability problems early on, the prototype of the CDSS was evaluated in two separate usability testing rounds with geriatric physicians. In the first round of usability testing, data saturation was reached after five physicians. The physicians (n = 5) identified a total of 74 usability problems with the CDSS, with a mean Nielsen's severity rating classification of 2.49 (i.e., between minor to major usability problems) [52]. Regarding major to severe usability problems, physicians perceived difficulties with the general navigability of the system (e.g., no "back button") and the medical terminology used within the system (e.g., what is meant by "lowest dose" or "minimal effective dose" for deprescribing medications). After these usability problems were solved, the CDSS underwent the second round of usability testing with physicians (n = 6, three of whom had not participated in the first round). Data saturation was reached after the inclusion of the sixth physician. The number of total usability problems identified by physicians was 11 and the mean Nielsen's severity rating was 1.87 (i.e., between cosmetic to minor problems) [52]. The most pressing usability problems were improved (e.g., fixing hyperlinks and task-orientated buttons), and the final version of the CDSS was developed. After each usability testing round, the user

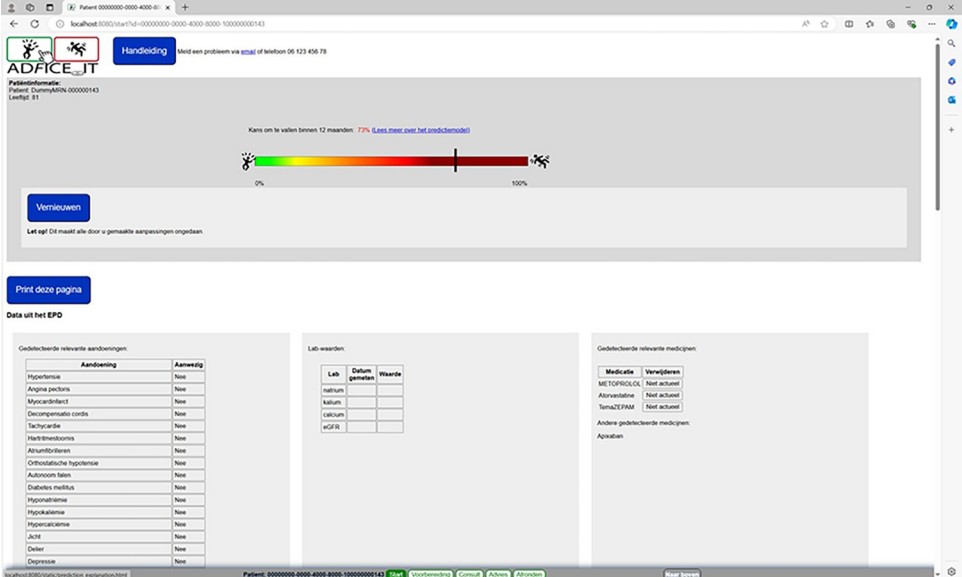

**Fig 1. Start page of the AD*F*ICE_IT CDSS.**

interface and clinical knowledge base of the CDSS were further optimized, which led to the development of the final CDSS.

**The final AD*F*ICE_IT CDSS.**   The source code for the AD*F*ICE_IT software is available at https://github.com/adfice-it. A connection between the EHR and the CDSS was made for the extraction of patient data into the CDSS. This data is used to provide patient-specific advice and to calculate a patient's personalized fall risk estimate. The final version of the CDSS is depicted in Figs 1–6. The CDSS consists of 5 web pages. Page 1 (titled "Start" in Fig 1) is the landing page of the CDSS. On this page the physician can (1) check whether relevant patient information (e.g., age, morbidities, list of medications) was correctly extracted from the EHR, and add missing data for calculating the fall risk estimate; (2) view a graphical representation of the patient's personalized fall risk estimate by means of a gradient scale; (3) view a model for shared decision-making that can be implemented during the consultation (in Fig 2); and (4) view a user guide of the system (titled "Handleiding"). Additionally, physicians have access to the patient identifier and personalized fall risk estimate during the entire consultation, as this information is displayed in the form of a horizontal menu bar on all subsequent pages.

Page 2 (titled "Preparation" in Fig 3) lists each medication taken by the patient and structurally provides the physician with patient-specific deprescribing advice for each listed medication in an attempt to facilitate optimal deprescribing for the physician. Page 2 also provides relevant non-medication related information for preventing falls in older patients (e.g., referral to fall prevention interventions, leaflets, etc.). The clinician can select the treatment options and/or non-medication related information they want to discuss with the patient. Moreover, hyperlinks to third-party sources are provided for additional information about the listed medications (i.e., "Farmacotherapeutisch Kompas") and related deprescribing advice (i.e., the guidelines used to formulate that advice).

Page 3 (titled "Consult" in Fig 4) provides an overview of the treatment options that the physician selected to discuss together with the patient (i.e., the deprescribing advice or leaflets selected in page 2). Based on the discussion with the patient, a final treatment plan is determined. Page 4 (titled "Advice" in Fig 5) displays the final treatment plan in a patient-friendly

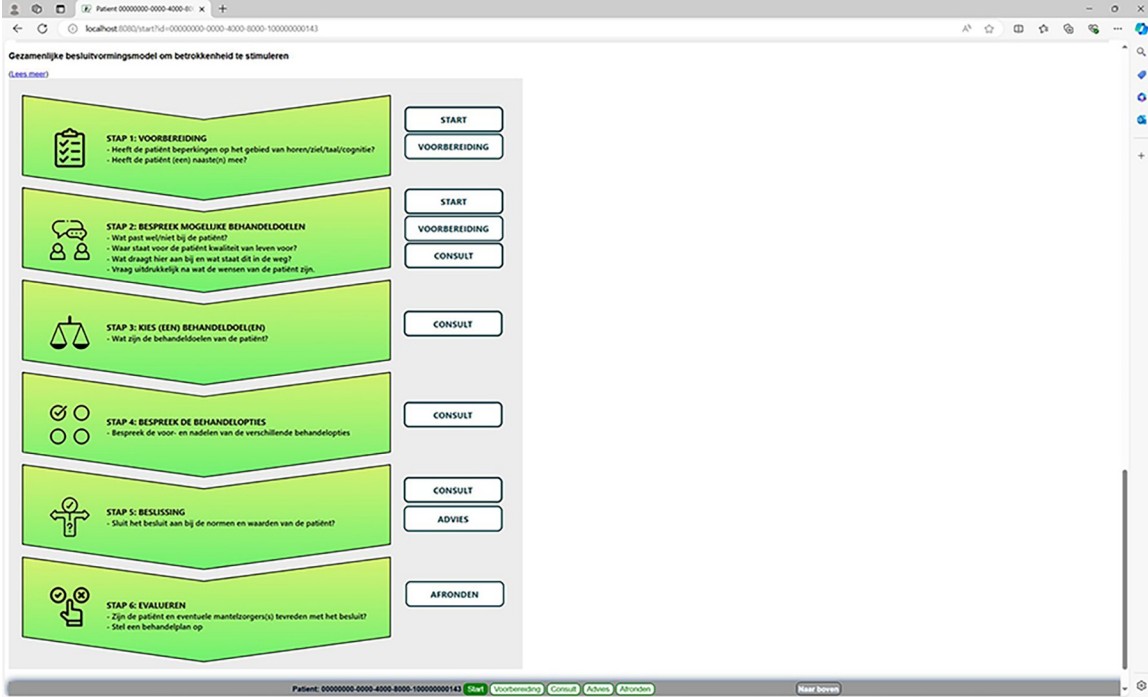

**Fig 2. Shared decision-making model of the AD*FICE*_IT CDSS.**

format, which is printable by the physician and accessible by the patient via the patient portal. In page 5 (titled "Wrap-up" in Fig 6), the physician is able to copy a summary of the consultation into the patient's EHR, for future storage.

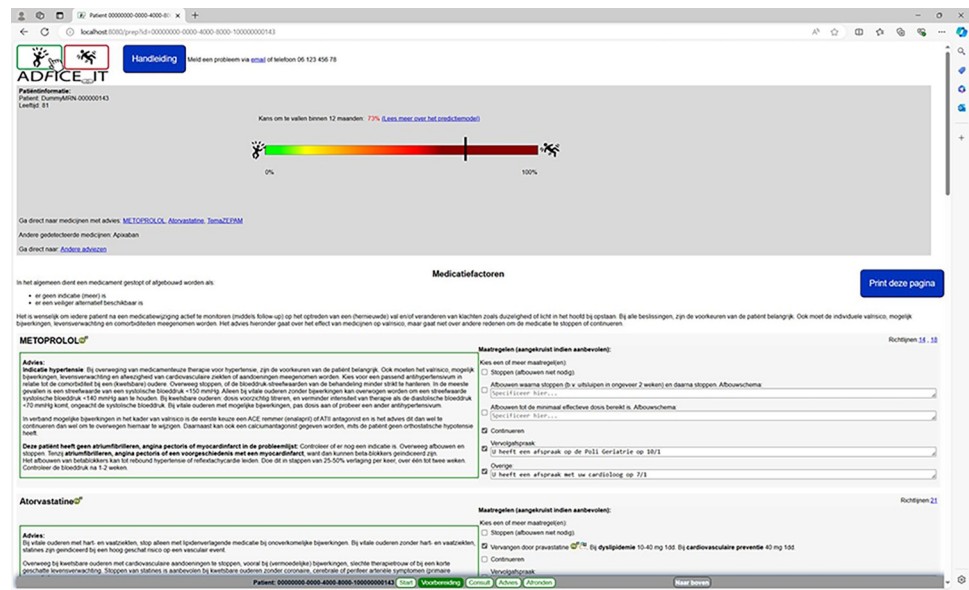

**Fig 3. Preparation page of the AD*FICE*_IT CDSS.**

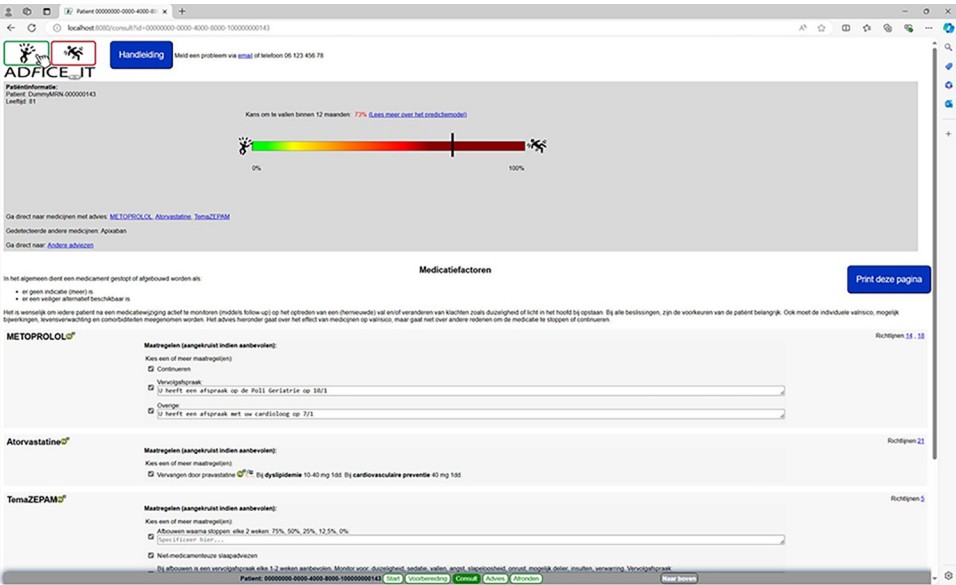

**Fig 4. Consult page of the AD*F*ICE_IT CDSS.**

## Discussion

This study outlined the user-centered development of a CDSS to support physicians in the optimum deprescribing of FRIDs in older adults (65+). The system was developed for the AD*F*ICE_IT intervention. In the development phase (phase I), a robust theoretical understanding about key components of the CDSS was cultivated. For this, both existing and new evidence was relied upon, in which collaboration with users of the system was sought throughout different stages of development. In phase I, a fall-risk prediction model was developed and internally validated. This prediction model was later integrated into the CDSS. Phase 1 also

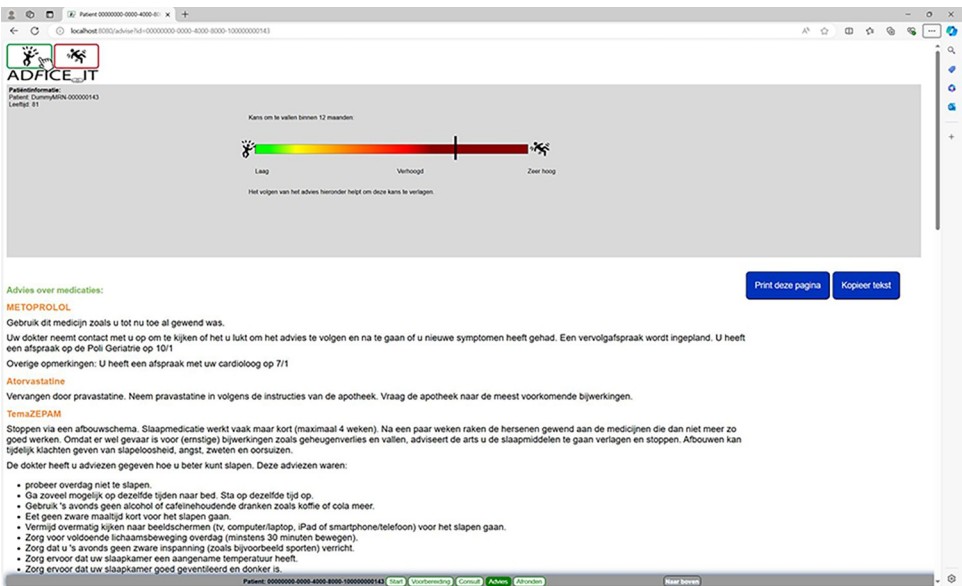

**Fig 5. Advice page of the AD*F*ICE_IT CDSS.**

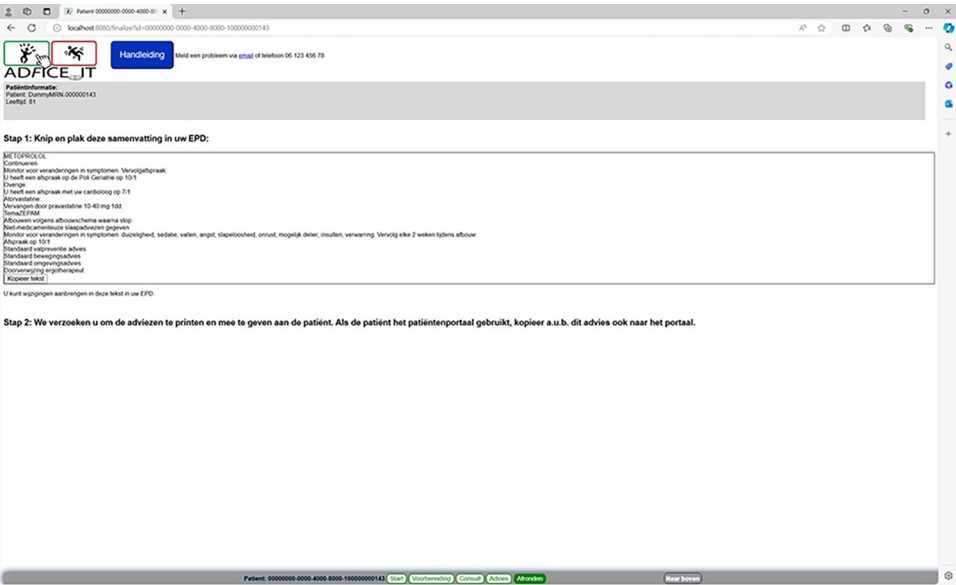

**Fig 6. Wrap-up page of the AD*F*ICE_IT CDSS.**

included the development of the knowledge, reasoning engine and software. Moreover, verification and validation testing were conducted to check for and subsequently correct errors relating to the generation of deprescribing advice. Together, the results from phase I guided the development of the first prototype of the CDSS. In the feasibility phase (phase II), two separate usability testing rounds with geriatric physicians were conducted to identify and address remaining usability issues within the CDSS that could hinder successful implementation of the system later on.

A key strength of this study was the ability to integrate the aggregated deprescribing guidelines and the fall-risk prediction model into the clinical knowledge base of our CDSS. For physicians, deprescribing FRIDs is an often complex task as differences in risk of adverse drug events make it hard to determine whether reducing a particular FRID or combination of FRIDs will result in a significant change in preventing a fall [12, 13]. According to Bloomfield et al., it is exactly this particular complexity in FRIDs deprescribing that has not been effectively addressed in past interventions [12]. This CDSS addresses this need by systematically guiding physicians through the deprescribing of one or more FRIDs. The system does this by first signaling a FRID from a patient's medication list, and subsequently leverages the stored deprescribing guidelines to advise physicians on how the identified FRID can be deprescribed optimally. Additionally, physicians can leverage the fall-risk prediction model as an adjunct to decision-making by assessing different advice on the basis of the patient's estimated risk of falling within a 12-month period.

Another key strength of our study was implementing a user-centered design approach to development. This, in our opinion, will greatly influence the acceptance of the system in its intended clinical care setting, as its functionalities and content are tailored to the needs and preferences of its users, namely geriatric physicians who treat older fall risk patients in hospitals. Past studies have shown low CDSS acceptance rates and a high number of usability barriers when physicians are not involved in the development process [24, 26–28]. Since the outcomes of the AD*F*ICE_IT intervention are dependent on physicians' use of the CDSS, insights from the system's users were gathered, at an early stage and throughout the

development process of the system, which led to several advantages. For example, the test-driven development allowed us to make changes to the code with confidence that earlier added functionality was not being disrupted. This also led to a more modular architecture within the code base, which facilitated making the changes suggested by users during the usability testing rounds. Additionally, while the decision to favor clarity over performance aided in this process, it should be noted that this does carry a limitation whereby the software–in its current form–is not scalable to support hundreds of concurrent users. Future versions of the software should consider enhancing scalability to support a higher number of concurrent users. Moreover, future development could improve testing of the client software further (e.g., unit and acceptance tests) by concentrating manipulation of the display to a few parts of the web page, which could allow for easier testing during development. While employing a rigorous verification and validation process led to an improved version of the CDSS after each feedback round, a limitation of this process was the involvement of one of the two clinicians during earlier verification and validation steps. This may have resulted in higher agreement, even though the system did perform well with the second clinician, who was not otherwise involved in the development process. Future versions of the CDSS should consider incorporating additional clinicians in the validation process to enhance robustness and, consequently, strengthen the acceptability and applicability of the CDSS recommendations among clinicians. Additionally, while the steps described in this paper closely align with the quality lifecycle model described in the ISO/IED 25010:2023 standard, standards such as ISO/IEC 25010 could help guide improvements to the software.

A final strength of our CDSS is that the system displays several different treatment options (i.e., advice) for each identified FRID (i.e., using guideline-based reasoning), which provides an opportunity for physicians to discuss preferred treatment options together with the patient. This can foster joint medication management, which could help in tackling the observed low rates of deprescribing compliance among patients [60]. In older patients, Van Weert et al. showed that shared decision-making was effective at enhancing both medication-related knowledge and adherence rates, which we believe could have a positive effect on the outcomes of the ADFICE_IT intervention as well [21]. Specifically, the effectiveness of our CDSS (along with the patient portal) is currently being tested in a multicenter, cluster-randomized controlled trial with process evaluation in several Dutch hospitals [61].

## Conclusions

This study followed a user-centered design approach to development to develop a CDSS intended for use in hospitals with older fall-risk patients in the Netherlands. The CDSS supports physicians in the optimum deprescribing of FRIDs to prevent falls in older adults (65+), leveraging aggregated deprescribing guidelines, a validated fall-risk prediction model, and shared decision-making. Moreover, due to the continued involvement of users, we were able to ensure that the CDSS was both useful and user-friendly as each new feedback round led to an improved version of the system. The CDSS (and patient portal) is currently being tested in a multicenter, cluster-randomized controlled trial with process evaluation at hospitals in the Netherlands (i.e., the ADFICE_IT intervention).

## Acknowledgments

The authors thank Tsvetan Yordanov, Rutger Bazen and Stephen Madson for their contributions to the software, and Leonie Westerbeek for her contribution to the user interface. The authors thank Eveline Poelgeest, Gerrit Jan Hafkamp, Irene Gomez Bruinewoud, Hester van

der Kroon, Hanna Willems, Suzanne Bleker, Oscar Smeekes and Stephanie van der Woude for their contributions to testing the feasibility of the AD*FICE*_IT CDSS.

## Author Contributions

**Conceptualization:** Natasja M. van Schoor, Julia C. M. van Weert, Nathalie van der Velde.

**Data curation:** Sara S. Groos, Kelly K. de Wildt, Bob van de Loo, Lotta J. Seppala, Kim J. Ploegmakers.

**Formal analysis:** Sara S. Groos, Kelly K. de Wildt.

**Funding acquisition:** Natasja M. van Schoor, Nathalie van der Velde.

**Methodology:** Sara S. Groos, Kelly K. de Wildt, Bob van de Loo, Stephanie Medlock, Lotta J. Seppala, Kim J. Ploegmakers.

**Project administration:** Sara S. Groos, Kelly K. de Wildt, Bob van de Loo, Annemiek J. Linn, Lotta J. Seppala, Kim J. Ploegmakers, Natasja M. van Schoor, Nathalie van der Velde.

**Software:** Stephanie Medlock, Kendrick M. Shaw, Eric K. Herman.

**Supervision:** Annemiek J. Linn, Stephanie Medlock, Natasja M. van Schoor, Julia C. M. van Weert, Nathalie van der Velde.

**Writing – original draft:** Sara S. Groos.

**Writing – review & editing:** Kelly K. de Wildt, Bob van de Loo, Annemiek J. Linn, Stephanie Medlock, Kendrick M. Shaw, Eric K. Herman, Lotta J. Seppala, Kim J. Ploegmakers, Natasja M. van Schoor, Julia C. M. van Weert, Nathalie van der Velde.

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
