## [Decision Letter · Decision Letter 0]

17 Jun 2024

PONE-D-23-43080Development of the ADFICE_IT clinical decision support system to assist deprescribing of fall-risk increasing drugs: A user-centered design approachPLOS ONE

Dear Dr. Groos,

Thank you for submitting your manuscript to PLOS ONE. After careful consideration, we feel that it has merit but does not fully meet PLOS ONE’s publication criteria as it currently stands. Therefore, we invite you to submit a revised version of the manuscript that addresses the points raised during the review process.

The main problme with the manuscript it mostly presents data that has already been published. However; numerous components covered in the manuscript is worth giving a try. Especially for clinically desicion supporting systems.

We look forward to receiving your revised manuscript.

Kind regards,

Mehmet Baysal

Academic Editor

PLOS ONE

Journal Requirements:

 [The ADFICE_IT project is supported by funding from the Netherlands Organization for Health Research and Development (ZonMw, Grant 848017004), The Hague and the Amsterdams Universiteitsfonds: Gepersonaliseerde Medicatieaanpassing bij Oudere Vallers. KMS was supported by US NIH grant T32-GM007592.

ZonMw: https://www.zonmw.nl/en

Amsterdams Universiteitsfonds: https://www.auf.nl/en

NIH: https://www.nih.gov/grants-funding ].  

4. We noted in your submission details that a portion of your manuscript may have been presented or published elsewhere. [This manuscript summarizes the (I) development and (II) feasibility phases of the ADFICE_IT CDSS which consisted of a scoping literature review, systematic literature review, European survey, and semi-structured interviews with physicians, including the aggregation and testing of deprescribing guidelines and the development of the fall-risk prediction model. In phase II, the feasibility of the ADFICE_IT CDSS was tested by means of two usability testing rounds. In this manuscript, we do not report the results

from the systematic literature review, European online survey, modified Delphi study, and development of the fall-risk prediction model as these have been published elsewhere by the research team. Instead, we outline how the aggregated results from these studies and others helped inform the final design/development requirements of the CDSS. Thus, we believe that this does not constitute dual publication.] Please clarify whether this publication was peer-reviewed and formally published. If this work was previously peer-reviewed and published, in the cover letter please provide the reason that this work does not constitute dual publication and should be included in the current manuscript.

Reviewers' comments:

Reviewer's Responses to Questions

**Comments to the Author**

1. Is the manuscript technically sound, and do the data support the conclusions?

Reviewer #1: Yes

Reviewer #2: Yes

Reviewer #3: Yes

2. Has the statistical analysis been performed appropriately and rigorously? 

Reviewer #1: Yes

Reviewer #2: Yes

Reviewer #3: Yes

3. Have the authors made all data underlying the findings in their manuscript fully available?

Reviewer #1: Yes

Reviewer #2: Yes

Reviewer #3: Yes

4. Is the manuscript presented in an intelligible fashion and written in standard English?

Reviewer #1: Yes

Reviewer #2: Yes

Reviewer #3: Yes

5. Review Comments to the Author

Reviewer #1: The manuscript describes the development of the ADFICE_IT clinical decision support system (CDSS) aimed at assisting physicians in deprescribing fall-risk increasing drugs (FRIDs) to reduce fall risk in older adults. The development process utilized a user-centered design approach involving systematic literature review, surveys, interviews, and usability testing with end users. Overall, the manuscript is a valuable contribution to the field of geriatric care and clinical decision support systems. It provides a robust framework for the development and implementation of similar tools in clinical practice. I only have a few minor questions and advices.

1. Introduction

The introduction provides a comprehensive overview of the importance of addressing fall risks in older adults and the potential of CDSS to support deprescribing practices. The background is well-researched, citing relevant studies that highlight the complexities of deprescribing FRIDs and the need for decision support tools.

The rationale for the study is clearly articulated.

The background information is well-supported by recent literature.

The objectives of the study are clearly stated, focusing on the development of a CDSS to support the deprescribing of FRIDs and its user-centered design process.

2. Methods

The methods section is detailed, describing a multi-phase approach that includes literature reviews, surveys, interviews, and usability testing according to the Medical Research Council framework. Comprehensive description of each phase of the development process is provided.

- Please explain what you mean with "Dutch fall clinics" as these are not common in all Eurpean countries.

- What was the profession of the physicians (geriatrics?)

- The sample size for the usability testing (n = 11) is relatively small, which might limit the generalizability of the findings.

- More detail on the selection criteria for survey and interview participants would enhance reproducibility.

- "harmonized dataset (n = 5722) of two Dutch cohorts and a German cohort study of community dwelling older adults (65+)" Please briefly explain and characterize both cohorts.

4. Results

The results are presented in a clear and structured manner, detailing the findings from literature reviews, surveys, interviews, and usability testing rounds.

- final prediction model consisted of the following 14 predictors: Briefly explain how these variables were measured (e.g. fear of falling, gait speed etc). Why did you enter both, one fall and two falls within last 12 months as independent variable in the model?

-prediction model: drugs for urinary frequency and incontinence: what exactly? Only drugs with central side effects or all drugs for urge incontinence?

- The presentation of results from the prediction model development could benefit from more detailed statistical analysis and validation metrics.

5. Discussion

The discussion is well-rounded, addressing the implications of the findings for clinical practice. It acknowledges the limitations of the study and suggests directions for future research.

- While limitations are acknowledged, there could be more emphasis on the potential biases introduced by the involvement of the same clinicians in both development and testing phases.

Reviewer #2: Thank you for the opportunity to review this manuscript.

This is an important contribution to the literature, giving an overview of the complete development and evaluation processes of a CDSS to assist deprescribing of fall-risk increasing drugs. The results of the multicenter, cluster randomized controlled trial with process evaluation at hospitals in the Netherlands will be of great interest.

My primary critique of this lengthy manuscript is that it provides primarily previously published data with very limited new information such as the results of the scoping review, presented in 11 lines (209-300) on page 15. Despite the appreciated summary in table 1, the many components described in the article required multiple back and forth between the methods and results sections.

A few specific questions / comments arose as I reviewed the manuscript:

P.4. line 71, the study registration number could be provided.

p.7 line 157, the patient portal is only mentioned here and briefly in the discussion. I suggest to either delete or develop further.

p.10, lines 190-191, the sentence ‘The scoping literature review assessed the risk communication needs of physicians.’ is the only methods information I found for this component. This is insufficient to describe the methods of a scoping review.

P.11, line 194, more information on the selection process of the participating physicians (GP, specialists, how they were selected…) is essential.

P.13, line 238, 10 patients seem minimal. Enough to cover the wide scope of recommendations?

p.13, line 259, will the system be used by other physicians such as GP? Could usability differ between GP and geriatric physicians?

p.14, lines 260-261, they were presented 3 cases but only completed at least one?

P.15, lines 289-300, I do not have enough methods information to appreciate the results of the scoping review.

P.15, lines 301-312, I do not have enough information. For example, the reader must go back to the abstract for the n=19. Who were those physicians? How were the interviews conducted? Was a standard questionnaire used?

p.16, line 314, how was this hierarchy decided?

p.16, lines 321-325, are all these variables (e.g. educational status, grip strength, fear of falling) readily available in a majority of EHR?

P.18, line 336, was ‘input from external expert developers’ received?

P.18, lines 343-345, what are the numbers? 95% and 100% agreement on what?

p. 18, line 348, ’74 individual usability problems’. What is the context? The three hypothetical cases? How many of those cases were evaluated by the 5 physicians?

P. 18, line 354, I have to go back to the abstract to find the n=11 and understand that these are 6 new physicians.

P.18, line 355, give the number of problems instead of the 16%.

p.19, lines 364, the figures were unreadable. Unfortunately, this portion of the manuscript could not be assessed.

Reviewer #3: The manuscript presents a technically sound piece of scientific research that describes the development and evaluation of the ADFICE_IT clinical decision support system (CDSS) designed to assist in the deprescribing of fall-risk increasing drugs (FRIDs) in older adults. The study is well-grounded, with data supporting the conclusions drawn. The experiments were conducted rigorously, with appropriate controls, replication, and sample sizes.

Strengths:

1. User-Centered Design Approach: The involvement of end users throughout the development process ensures that the CDSS is practical, useful, and likely to be accepted in clinical settings. This approach has been shown to improve technology adoption by healthcare professionals.

2. Integration of Deprescribing Guidelines and Risk Prediction Models: The CDSS leverages a combination of deprescribing guidelines and a validated fall-risk prediction model, which addresses the complexity involved in deprescribing FRIDs. This systematic approach can significantly aid physicians in making informed decisions to reduce fall risks in older adults.

3. Iterative Development and Usability Testing: The use of agile methodologies and iterative feedback from usability testing rounds allowed the authors to refine the CDSS continuously. This resulted in a user-friendly system with minimized usability barriers, enhancing its effectiveness and efficiency in clinical practice.

4. Detailed Validation Process: The CDSS underwent thorough verification and validation testing, including both unit tests and real patient data validation. This rigorous process ensured the reliability and accuracy of the system's recommendations.

Areas for Improvement:

1. Scalability: Although clarity of code was prioritized over efficiency and scalability, future versions of the software should consider enhancing scalability to support a higher number of concurrent users. This will be particularly important for broader implementation across multiple clinical settings.

2. Broader Validation: While the involvement of one clinician in the early validation steps was beneficial, incorporating a more diverse group of clinicians in the initial stages could further strengthen the robustness of the validation process. This would ensure that the CDSS recommendations are universally acceptable and applicable.

3. Comprehensive Documentation: Providing more detailed documentation and user guides could facilitate easier implementation and training for new users. This would help in overcoming initial barriers to adoption and ensure that users can fully utilize the system's capabilities.

Overall Impression:

The manuscript is well-written and presented in clear, standard English. The data underlying the findings are fully available, adhering to PLOS ONE's data availability policy. The development and evaluation of the ADFICE_IT CDSS represent a significant advancement in supporting the safe deprescribing of FRIDs to prevent falls in older adults. The user-centered design approach and rigorous testing processes enhance the system's practicality and reliability.

The manuscript's clear and systematic presentation, combined with the robustness of the research methods, makes it a valuable contribution to the field. It addresses a critical aspect of geriatric care, and its findings have the potential to improve patient outcomes significantly.

Recommendation: Minor revision

Reasoning:

The manuscript presents a significant and well-executed study on the development and evaluation of the ADFICE_IT clinical decision support system (CDSS) for assisting in the deprescribing of fall-risk increasing drugs (FRIDs) in older adults. The study is technically sound, with robust data supporting the conclusions, and it follows rigorous experimental protocols. The user-centered design approach and iterative usability testing have resulted in a practical and user-friendly system that addresses a critical aspect of geriatric care.

However, there are a few areas that could benefit from further improvement before publication:

Scalability: The manuscript should address potential scalability issues and discuss plans for future enhancements to support a higher number of concurrent users.

Comprehensive Documentation: Providing more detailed documentation and user guides will facilitate easier implementation and training, helping to overcome initial adoption barriers.

Alignment with Standars: Consider the pertinence of describing in the manuscript how the ADFICE_IT CDSS aligns with the quality characteristics defined in ISO/IEC 25010:2011. Including this information can strengthen the manuscript by demonstrating that the CDSS meets recognized benchmarks for software quality, thus enhancing its credibility and reliability. If the authors can access this standard, it is recommended to review it to ensure comprehensive alignment.

Addressing these minor issues will enhance the manuscript's clarity and ensure that the CDSS is well-positioned for broader implementation and use.

6. PLOS authors have the option to publish the peer review history of their article (what does this mean?). If published, this will include your full peer review and any attached files.

Reviewer #1: No

Reviewer #2: No

Reviewer #3: No

---

## [Author Response · Author response to Decision Letter 0]

8 Aug 2024

We express our gratitude to the reviewers for their insightful feedback, which has been invaluable in enhancing the quality of our manuscript. In an attached file ("Response to Reviewers") you will find our responses (in bold) to each reviewer’s comment together with references to relevant changes we made to the manuscript. The full edited manuscript with tracked changes is attached as well.

---

## [Decision Letter · Decision Letter 1]

22 Aug 2024

Development of the ADFICE_IT clinical decision support system to assist deprescribing of fall-risk increasing drugs: A user-centered design approach

PONE-D-23-43080R1

Dear Dr. Groos,

We’re pleased to inform you that your manuscript has been judged scientifically suitable for publication and will be formally accepted for publication once it meets all outstanding technical requirements.

Kind regards,

Mehmet Baysal

Academic Editor

PLOS ONE

Additional Editor Comments (optional):

Reviewers' comments:

Reviewer's Responses to Questions

**Comments to the Author**

1. If the authors have adequately addressed your comments raised in a previous round of review and you feel that this manuscript is now acceptable for publication, you may indicate that here to bypass the “Comments to the Author” section, enter your conflict of interest statement in the “Confidential to Editor” section, and submit your "Accept" recommendation.

Reviewer #3: All comments have been addressed

2. Is the manuscript technically sound, and do the data support the conclusions?

Reviewer #3: Yes

3. Has the statistical analysis been performed appropriately and rigorously? 

Reviewer #3: Yes

4. Have the authors made all data underlying the findings in their manuscript fully available?

Reviewer #3: Yes

5. Is the manuscript presented in an intelligible fashion and written in standard English?

Reviewer #3: Yes

6. Review Comments to the Author

Reviewer #3: The manuscript presents a significant and well-conducted study on the development of a clinical decision support system (CDSS) aimed at optimizing the deprescribing of fall-risk increasing drugs in older adults. The authors have thoroughly addressed the reviewers' comments, resulting in a manuscript that is technically sound and methodologically rigorous.

The authors acknowledge some limitations in the current version of the software, particularly regarding scalability and the breadth of clinical validation. While the software is functional, they recognize the need for future improvements to support a higher number of concurrent users and to expand validation to include a more diverse group of clinicians, which will enhance the robustness and generalizability of the system's recommendations. Additionally, the authors mention the potential alignment with recognized standards such as ISO/IEC 25010 in future versions, which could further strengthen the system's credibility and reliability.

Overall, the manuscript is a valuable contribution to the field, and with these forward-looking plans, it will be well-positioned for publication.

7. PLOS authors have the option to publish the peer review history of their article (what does this mean?). If published, this will include your full peer review and any attached files.

Reviewer #3: No

---

## [Editor Report · Acceptance letter]

26 Aug 2024

PONE-D-23-43080R1 

PLOS ONE

Dear Dr. Groos, 

I'm pleased to inform you that your manuscript has been deemed suitable for publication in PLOS ONE. Congratulations! Your manuscript is now being handed over to our production team.

Kind regards, 

on behalf of

Dr. Mehmet Baysal 

Academic Editor

PLOS ONE